# Metadata Archaeology: Unearthing Data Subsets by Leveraging Training Dynamics

**Shoaib Ahmed Siddiqui**
University of Cambridge
msas3@cam.ac.uk

**Nitarshan Rajkumar**
University of Cambridge
nr500@cam.ac.uk

**Tegan Maharaj**
University of Toronto
tegan.maharaj@utoronto.ca

**David Krueger**
University of Cambridge
dsk30@cam.ac.uk

**Sara Hooker**
Cohere for AI
sarahooker@cohere.com

## Abstract

Modern machine learning research relies on relatively few carefully curated datasets. Even in these datasets, and typically in 'untidy' or raw data, practitioners are faced with significant issues of data quality and diversity which can be prohibitively labor intensive to address. Existing methods for dealing with these challenges tend to make strong assumptions about the particular issues at play, and often require *a priori* knowledge or metadata such as domain labels. Our work is orthogonal to these methods: we instead focus on providing a unified and efficient framework for *Metadata Archaeology* – uncovering and inferring metadata of examples in a dataset. We curate different subsets of data that might exist in a dataset (e.g. mislabeled, atypical, or out-of-distribution examples) using simple transformations, and leverage differences in learning dynamics between these **probe suites** to infer metadata of interest. Our method is on par with far more sophisticated mitigation methods across different tasks: identifying and correcting mislabeled examples, classifying minority-group samples, prioritizing points relevant for training and enabling scalable human auditing of relevant examples.

## 1 Introduction

Modern machine learning is characterized by ever-larger datasets and models. The expanding scale has produced impressive progress (Wei et al., 2022; Kaplan et al., 2020; Roberts et al., 2020) yet presents both optimization and auditing challenges. Real-world dataset collection techniques often result in significant label noise (Vasudevan et al., 2022), and can present significant numbers of redundant, corrupted, or duplicate inputs (Carlini et al., 2022). Scaling the size of our datasets makes detailed human analysis and auditing labor-intensive, and often simply infeasible. These realities motivate a consideration of how to efficiently characterize different aspects of the data distribution.

Prior work has developed a rough taxonomy of data properties, or **metadata** which different examples might exhibit, including but not limited to: *noisy* (Wu et al., 2020; Yi and Wu, 2019; Thulasidasan et al., 2019a;b), *atypical* (Hooker et al., 2020; Buolamwini and Gebru, 2018; Hashimoto et al., 2018; Słowik and Bottou, 2021), *challenging* (Ahia et al., 2021; Baldock et al., 2021; Paul et al., 2021; Agarwal et al., 2021), *prototypical or core subset selection* (Paul et al., 2021; Sener and Savarese, 2018; Shim et al., 2021; Huggins et al., 2017; Sorscher et al., 2022) and *out-of-distribution* (Hendrycks et al., 2019; LeBrun et al., 2022). While important progress has been made on some of these metadata categories individually, these categories are typically addressed in isolation reflecting an overly strong assumption that only one, known issue is at play in a given dataset.

For example, considerable work has focused on the issue of label noise. A simple yet widely-used approach to mitigate label noise is to remove the impacted data examples (Pleiss et al., 2020). However, it has been shown that it is challenging to distinguish difficult examples

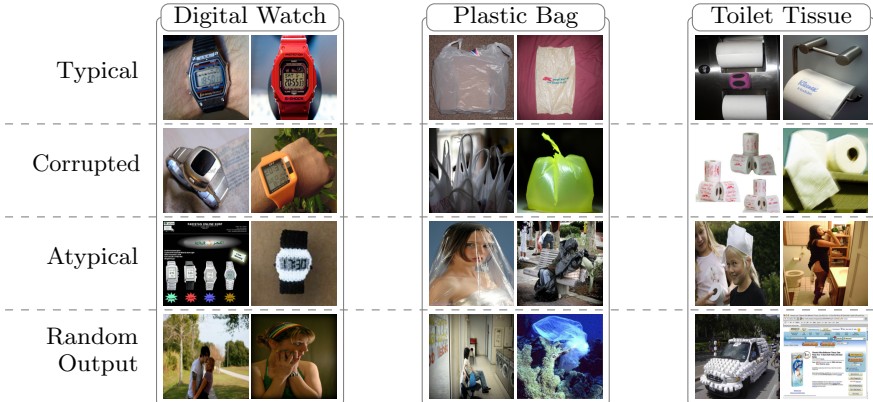

Figure 1: Examples surfaced through the use of *MAP-D* on ImageNet train set. **Column** title is the ground truth class, **row** title is the metadata category assigned by *MAP-D*. *MAP-D* performs metadata archaeology by curating a probe set and then probing for similar examples based on training dynamics. This approach can bring to light biases, mislabelled examples, and other dataset issues.

from noisy ones, which often leads to useful data being thrown away when both noisy and atypical examples are present (Wang et al., 2018; Talukdar et al., 2021).

Meanwhile, loss-based prioritization (Jiang et al., 2019; Katharopoulos and Fleuret, 2018) techniques essentially do the opposite – these techniques *upweight* high loss examples, assuming these examples are challenging yet learnable. These methods have been shown to quickly degrade in performance in the presence of even small amounts of noise since upweighting noisy samples hurts generalization (Hu et al., 2021; Paul et al., 2021).

The underlying issue with such approaches is the assumption of a single, known type of data issue. Interventions are often structured to identify examples as simple vs. challenging, clean vs. noisy, typical vs. atypical, in-distribution vs. out-of-distribution etc. However, large scale datasets may present subsets with many different properties. In these settings, understanding the interactions between an intervention and many different subsets of interest can help prevent points of failure. Moreover, relaxing the notion that all these properties are treated independently allows us to capture realistic scenarios where multiple metadata annotations can apply to the same datapoint. For example, a *challenging* example may be so because it is *atypical*.

In this work, we are interested in moving away from a siloed treatment of different data properties. We use the term **Metadata Archaeology** to describe the problem of inferring metadata across a more complete data taxonomy. Our approach, which we term **Metadata Archaeology via Probe Dynamics (*MAP-D*)**, leverages distinct differences in training dynamics for different curated subsets to enable specialized treatment and effective labelling of different metadata categories. Our methods of constructing these probes are general enough that the same probe category can be crafted efficiently for many different datasets with limited domain-specific knowledge.

We present consistent results across six image classification datasets, CIFAR-10/CIFAR-100 (Krizhevsky et al., 2009), ImageNet (Deng et al., 2009), Waterbirds (Sagawa et al., 2020), CelebA (Liu et al., 2015) , Clothing1M (Xiao et al., 2015) and two models from the ResNet family (He et al., 2016). Our simple approach is competitive with far more complex mitigation techniques designed to only treat one type of metadata in isolation. We summarize our contributions as:

- We propose **Metadata Archaeology**, a unifying and general framework for uncovering latent metadata categories.

- We introduce and validate the approach of **Metadata Archaeology via Probe Dynamics (*MAP-D*)**: leveraging the training dynamics of curated data subsets called **probe suites** to infer other examples' metadata.

- We show how *MAP-D* could be leveraged to audit large-scale datasets or debug model training, with negligible added cost - see Figure 1. This is in contrast to prior work which presents a siloed treatment of different data properties.
- We use *MAP-D* to identify and correct mislabeled examples in a dataset. Despite its simplicity, *MAP-D* is on-par with far more sophisticated methods, while enabling natural extension to an arbitrary number of modes.
- Finally, we show how to use *MAP-D* to identify minority group samples, or surface examples for data-efficient prioritized training.

## 2  Metadata Archaeology via Probe Dynamics (MAP-D)

Metadata is data about data, for instance specifying when, where, or how an example was collected. This could include the provenance of the data, or information about its quality (e.g. indicating that it has been corrupted by some form of noise). An important distinguishing characteristic of metadata is that it can be *relational*, explaining how an example compares to others. For instance, whether an example is typical or atypical, belongs to a minority class, or is out-of-distribution (OOD), are all dependent on the entire data distribution.

The problem of **metadata archaeology** is the inference of metadata $m \subset \mathcal{M}$ given a dataset $\mathcal{D}$. While methods for inferring $m$ might also be useful for semi-supervised labelling or more traditional feature engineering, and vice versa, it is the relational nature of metadata that makes this problem unique and often computationally expensive.

### 2.1  Methodology

**Metadata Archaeology via Probe Dynamics (*MAP-D*)**, leverages differences in the statistics of learning curves across metadata features to infer the metadata of previously unseen examples.

We consider a model which learns a function $f_\theta : \mathcal{X} \mapsto \mathcal{Y}$ with trainable weights $\theta$. Given the training dataset $\mathcal{D}$, $f_\theta$ optimizes a set of weights $\theta^*$ by minimizing an objective function $L$ with loss $l$ for each example. We assume that the learner has access to two types of samples for training. *First* is a training set $\mathcal{D}$:

$$\mathcal{D} := \big\{ (x_1, y_1), \ldots, (x_N, y_N) \big\} \subset \mathcal{X} \times \mathcal{Y} \ , \tag{1}$$

where $\mathcal{X}$ represents the data space and $\mathcal{Y}$ the set of outcomes associated with the respective instances. Examples in the data space are also assumed to have associated, but unobserved, metadata $m \subset \mathcal{M}$. *Secondly*, we assume the learner to also have access to a small curated subset of $j$ samples ($j \leq N$) associated with metadata $m \subset \mathcal{M}$, i.e.:

$$\mathcal{D}_m := \{ (x_1, y_1, m_1), \ldots, (x_j, y_j, m_j) \} \subset \mathcal{X} \times \mathcal{Y} \times \mathcal{M} \tag{2}$$

We refer to these curated subsets as probe suites. A key criteria is for our method to require very few annotated probe examples ($j \ll N$). In this work, we focus on probe suits which can be constructed algorithmically, as human annotations of metadata require costly human effort to maintain.

### 2.1.1  Assigning Metadata Features to Unseen Examples

*MAP-D* works by comparing the performance of a given example to the learning curves typical of a given probe type. Our approach is motivated by the observation that different types of examples often exhibit very different learning dynamics over the course of training (see Figure 3). In an empirical risk minimization setting, we minimize the average training loss across all training points.

$$L_{(\theta)} = \frac{1}{N} \sum_{i=1}^{N} \ell\,(x_i, y_i; \theta)$$

However, performance on a given subset will differ from the average error. Specifically, we firstly evaluate the learning curves of individual examples:

$$\mathbf{s}_i^t := (\ell(x_i, y_i; \theta_1), \ell(x_i, y_i; \theta_2), \ldots, \ell(x_i, y_i; \theta_t) \mid (x_i, y_i) \in \mathcal{D}) \tag{3}$$

| Black bear | Dishwasher | School bus | Mud turtle | Jeep | Loafer |
|---|---|---|---|---|---|

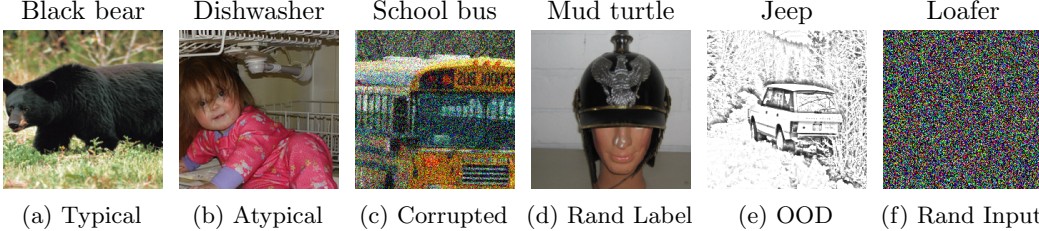

| (a) Typical | (b) Atypical | (c) Corrupted | (d) Rand Label | (e) OOD | (f) Rand Input |
|---|---|---|---|---|---|

Figure 2: An illustration of samples from our curated probes. Creating our probe suites requires no human annotation. We curate different subsets of data that might exist in a dataset including (a) typical, (b) atypical, (c) corrupted, (d) mislabeled, (e) out-of-distribution, and (f) random input examples, using simple transformations or existing scoring mechanisms.

where $\mathbf{s}_i^t$ denotes the learning curve for the $i^{\text{th}}$ training example, and $t$ is the current epoch[1]. An important property of the loss function is that it naturally integrates information regarding the whole data distribution via training. This is essential for metadata inference which is relational in nature. We then track the per-example performance on probes $\mathbf{g}$ for each metadata category $m \in \{m_1, \ldots, m_{|\mathcal{M}|}\}$, and refer to each probe as $\mathbf{g}(m)$.

$$\mathbf{g}_j^t(m) := (\ell(x_j, y_j; \theta_1), \ell(x_j, y_j; \theta_2), ..., \ell(x_j, y_j; \theta_t) \mid (x_j, y_j) \in \mathcal{D}_m) \tag{4}$$

where $\mathbf{g}_j^t(m)$ denotes the learning curve computed on the $j^{\text{th}}$ example chosen from a given probe category $m$. We use $\mathcal{D}_g$ as shorthand to refer to the set of all these trajectories for the different probe categories along with the category identity.

$$\mathcal{D}_\mathbf{g} := \Big((\mathbf{g}_1^t(m_1), m_1), \ldots, (\mathbf{g}_{|m_1|}^t(m_1), m_1), (\mathbf{g}_1^t(m_2), m_2), \ldots, (\mathbf{g}_{|m_{|\mathcal{M}|}|}^t(m_{|\mathcal{M}|}), m_{|\mathcal{M}|})\Big) \tag{5}$$

where $|m_c|$ refers the number of examples belonging to the metadata category $m_c$.

We assign metadata features to an unseen data point by looking up the example's nearest neighbour from $\mathcal{D}_\mathbf{g}$, using the Euclidean distance. In general, assignment of probe type could be done via any classification algorithm. However, in this work we use $k$-NN ($k$-Nearest Neighbours) for its simplicity, interpretability and the ability to compute the probability of multiple different metadata features.

$$p(m \mid \mathbf{s}_i^t) = \frac{1}{k} \sum_{(\mathbf{g}, \hat{m}) \,\in\, \text{NN}(\mathbf{s}_i^t, \mathcal{D}_\mathbf{g}, k)} \mathbb{1}_{\hat{m}=m} \tag{6}$$

where $p(m \mid \mathbf{s}_i^t)$ is the probability assigned to probe category $m$ based on the $k$ nearest neighbors for learning curve of the $i^{\text{th}}$ training example from the dataset, and $\text{NN}(\mathbf{s}_i^t, \mathcal{D}_g, k)$ represents the top-$k$ nearest neighbors for $\mathbf{s}_i^t$ from $\mathcal{D}_\mathbf{g}$ (probe trajectory dataset) based on Euclidean distance between the loss trajectories for all the probe examples and the given training example. We fix $k=20$ in all our experiments.

This distribution over probes (i.e metadata features) may be of primary interest, but we are sometimes also interested in seeing which metadata feature a given example most strongly corresponds to; in this case, we compute the argmax:

$$m_i' = \arg\max_{m \in \mathcal{M}} \; p(m \mid \mathbf{s}_i^t) \tag{7}$$

where $m_i'$ denotes the assignment of the $i^{\text{th}}$ example to a particular probe category.

We include the probe examples in the training set unless specified otherwise; excluding them in training can result in a drift in trajectories, and including them allows tracking of training dynamics.

---

[1] A coarser or finer resolution for the learning curves could also be used, e.g. every $n$ steps or epochs. All experiments in this work use differences computed at the end of the epoch.

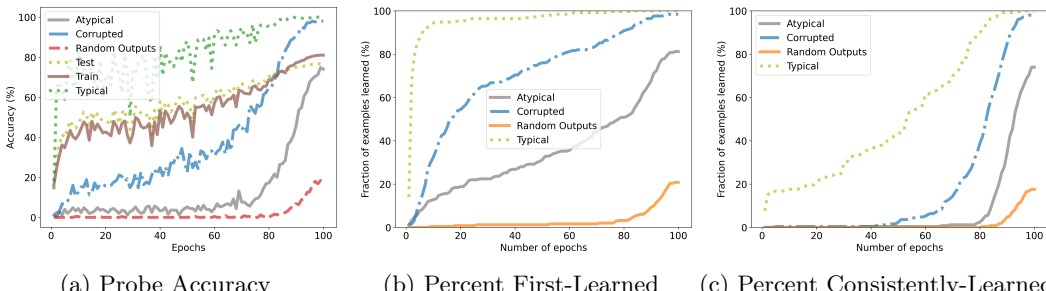

| (a) Probe Accuracy | (b) Percent First-Learned | (c) Percent Consistently-Learned |

Figure 3: Probe categories are distinguishable via learning dynamics of a ResNet-50 trained on ImageNet, validating the approach of *MAP-D*. For each of the probe categories and at each epoch, we plot **(a)** each probe's average accuracy; **(b)** the cumulative fraction of examples once predicted correctly by the nth epoch; and **(c)** the fraction that remain predicted correctly on all subsequent epochs.

## 2.2 Probe Suite Curation

While probe suites can be constructed using human annotations, this can be very expensive to annotate (Andrus et al., 2021; Veale and Binns, 2017). In many situations where auditing is desirable (e.g. toxic or unsafe content screening), extensive human labour is undesirable or even unethical (Steiger et al., 2021; Shmueli et al., 2021). Hence, in this work, we focus on probes that can be computationally constructed for arbitrary datasets – largely by using simple transformations and little domain-specific knowledge. We emphasize that our probe suite is not meant to be exhaustive, but to provide enough variety in metadata features to demonstrate the merits of metadata archaeology.

We visualize these probes in Figure 2, and describe below:

1. **Typical** We quantify typicality by thresholding samples with the top consistency scores from Jiang et al. (2020) across all datasets. The consistency score is a measure of expected classification performance on a held-out instance given training sets of varying size sampled from the training distribution.
2. **Atypical** Similarly, atypicality is quantified as samples with the lowest consistency scores from Jiang et al. (2020).
3. **Random Labels** Examples in this probe have their labels replaced with uniform random labels, modelling label noise.
4. **Random Inputs & Labels** These noisy probes are comprised of uniform $\mathcal{U}(0,1)$ noise sampled independently for every dimension of the input. We also randomly assign labels to these samples.
5. **Corrupted Inputs** Corrupted examples are constructed by adding Gaussian noise with 0 mean and 0.1 standard deviation for CIFAR-10/100 and 0.25 standard deviation for ImageNet. These values were chosen to make the inputs as noisy as possible while still being (mostly) recognizable to humans.

We curate 250 training examples for each probe category. For categories other than Typical/Atypical, we sample examples at random and then apply the corresponding transformations. We also curate 250 test examples for each probe category to evaluate the accuracy of our nearest neighbor assignment of metadata to unseen data points, where we know the true underlying metadata.

**Training Details and Architectures:** We list all the training details and architectures in Appendix A. All our experiments are based on ResNet-50 (He et al., 2016), except label correction experiments which are based on ResNet-18 following Arazo et al. (2019).

## 3 Experiments and Discussion

In the following sections, we perform experiments across 6 datasets: CIFAR-10/100, ImageNet, Waterbirds, CelebA, and Clothing1M. For details regarding the experimental setup,

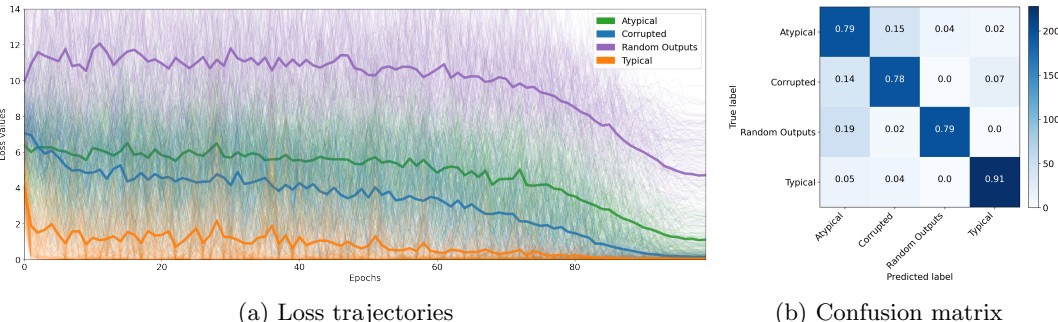

(a) Loss trajectories             (b) Confusion matrix

Figure 4: Sanity check showing performance of *MAP-D* on the probe suite test set with four main probe categories using ResNet-50 on ImageNet, where we know the ground-truth metadata. **(a)** Solid line shows the mean learning curve while translucent lines are randomly sampled 250 individual trajectories for each probe category. Again, the separation of different probes is evident both in the dynamics over the course of training. **(b)** show confusion matrices between the true vs. predicted metadata features, demonstrating strong performance of the probes (see Fig. 12 for depiction of performance on all probe categories).

see Appendix A. We first evaluate **convergence dynamics** of different probe suites (Section 3.1), validating the approach of *MAP-D*. We then qualitatively demonstrate the ability to **audit datasets** using *MAP-D* (Section 3.2), and evaluate performance on a variety of downstream tasks: **noise correction** (Section 3.3), **prioritizing points for training** (Section 3.4), and **identifying minority-group samples** (Section 3.5).

## 3.1 Probe Suite Convergence Dynamics

In Figure 3, we present the training dynamics on the probe suites given a ResNet-50 model on ImageNet. For all datasets, we observe that probe suites have distinct learning convergence trajectories, demonstrating the efficacy of leveraging differences in training dynamics for the identification of probe categories. We plot average 1) **Probe Accuracy** over the course of training, 2) the **Percent First-Learned** i.e. the percentage of samples which have been correctly classified once (even if that sample was be later forgotten) over the course of training, and 3) the **Percent Consistently-Learned** i.e. the percentage of samples which have been learned and will not be forgotten for the rest of training.

We observe consistent results across all dimensions. Across datasets, the `Typical` probe has the fastest rate of learning, whereas the `Random Outputs` probe has the slowest. When looking at *Percent First-Learned* in Figure 3, we see a very clear natural sorting by the difficulty of different probes, where natural examples are learned earlier as compared to corrupted examples with synthetic noise. Examples with random outputs are the hardest for the model.

We also observe that probe ranking in terms of both *Percent First-Learned* and *Percent Consistently-Learned* is stable across training, indicating that model dynamics can be leveraged consistently as a stable signal to distinguish between different subsets of the distribution at any point in training. These results motivate our use of learning curves as signal to infer unseen metadata.

## 3.2 Auditing Datasets

A key motivation of our work is that the large size of modern datasets means only a small fraction of datapoints can be economically inspected by humans. In safety-critical or otherwise sensitive domains such as healthcare diagnostics (Xie et al., 2019; Gruetzemacher et al., 2018; Badgeley et al., 2019; Oakden-Rayner et al., 2019), self-driving cars (NHTSA, 2017), hiring (Dastin, 2018; Harwell, 2019), and many others, providing tools for domain experts to audit models is of great importance to ensure scalable oversight.

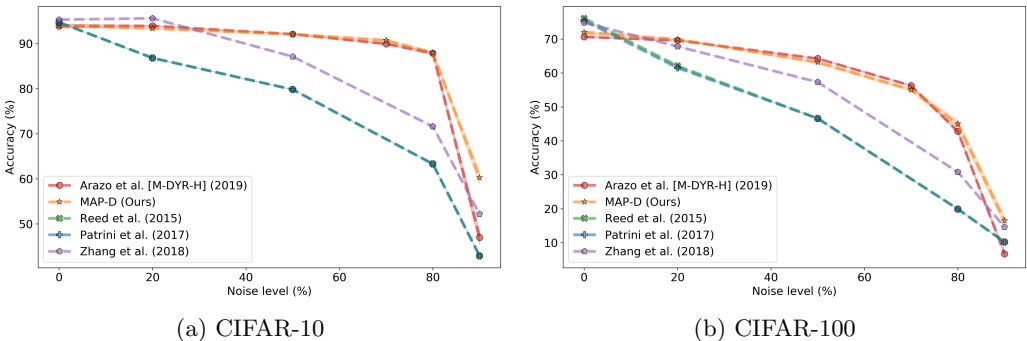

(a) CIFAR-10                              (b) CIFAR-100

Figure 5: Comparison of different noise correction methods under the presence of label noise. Mean and standard deviation reported over 3 random runs. *MAP-D* is competitive with most other methods, many of which are particularly targeted towards this problem.

We apply *MAP-D* to infer the metadata features of the underlying dataset. In Fig. 1, we visualize class specific examples surfaced by *MAP-D* on the ImageNet train set. Our visualization shows that *MAP-D* helps to disambiguate effectively between different types of examples and can be used to narrow down the set of datapoints to prioritize for inspection. We observe clear semantic differences between the sets. In Fig. 1, we observe that examples surfaced as `Typical` are mostly well-centered images with a typical color scheme, where the only object in the image is the object of interest. Examples surfaced as `Atypical` present the object in unusual settings or vantage points, or feature differences in color scheme from the typical variants. We observe examples that would be hard for a human to classify using the `Random Output` probe category. For example, we see incorrectly labeled images of a digital watch, images where the labeled object is hardly visible, artistic and ambiguous images, and multi-object examples where several different labels may be appropriate. We visualize more examples from the `Random Output` probe category in Fig. 8.

As a sanity check, we also evaluate the performance of *MAP-D* on the held-out probe test set in Fig. 4, where we know the true underlying metadata used to curate that example. In particular, we compute performance on the four probes which are most easily separable via learning curves, and find that model was able to achieve high detection performance ($\sim 81.9\%$ accuracy). Results with all probe categories are presented in appendix E. The primary aim here is to demonstrate that *MAP-D* is sufficiently capable in detecting the defined probe categories in the dataset.

### 3.3 LABEL NOISE CORRECTION

Here we apply *MAP-D* to detect and correct label noise, a data quality issue that has been heavily studied in prior works (Zhang et al., 2017; Arazo et al., 2019; Arpit et al., 2017). We benchmark against a series of different baselines under the assumption of uniform label noise (Arazo et al., 2019; Zhang et al., 2017; Patrini et al., 2017; Reed et al., 2014), some of which are specifically developed to deal with label noise. We emphasize that our aim is not to develop a specialized technique for dealing with label noise, but to showcase that *MAP-D*, a general solution for metadata archaeology, also performs well on specialized tasks such as label correction.

To distinguish between clean and noisy samples using *MAP-D*, we add an additional *random sample* probe curated via a random sample from the (unmodified) underlying data, as a proxy for clean data. For this comparison, we follow the same experimental protocol as Arazo et al. (2019), where all the methods we benchmark against are evaluated.

Concretely, for any label correction scheme, the actual label used for training is a convex combination of the original label and the model's prediction based on the probability of the sample being either clean or noisy. Considering one-hot vectors, the correct label can be represented as:

$$\bar{y}_i = p(\texttt{clean} \mid \mathbf{s}_i^t) \times y_i + p(\texttt{noisy} \mid \mathbf{s}_i^t) \times \hat{y}_i \tag{8}$$

where $\bar{y}_i$ represents the corrected label used to train the model, $y_i$ represents the label present in the dataset weighted by the probability of the sample being clean $p(\texttt{clean} \mid \mathbf{s}_i^t)$, and $\hat{y}_i$ represents the model's prediction (a one-hot vector computed via argmax rather than predicted probabilities) weighted by the probability of the sample being noisy $p(\texttt{noisy} \mid \mathbf{s}_i^t)$. Since we are only considering two classes, $p(\texttt{clean} \mid \mathbf{s}_i^t) = 1 - p(\texttt{noisy} \mid \mathbf{s}_i^t)$. We employ the online *MAP-D* trajectory scheme in this case, where the learning curve is computed given all prior epochs completed as of that point. We use the actual probability estimate returned by MAP-D. We highlight the utility of these probability estimates by comparing against the binary prediction baseline in appendix C.

Despite the relative simplicity and generality of *MAP-D*, it generally performs as well as highly-engineered methods developed specifically for this task. Our results are presented in Fig. 5. Specifically, at extremely **high levels of noise**, *MAP-D* performs significantly better on both CIFAR-10 and CIFAR-100 in comparison to Arazo et al. (2019) (CIFAR-10: $\sim 47\%$ vs $\sim 59\%$; CIFAR-100: $\sim 6.5\%$ vs $\sim 16.5\%$).

We also show that *MAP-D* is robust against changes in the training setup, while Arazo et al. Arazo et al. (2019) struggles in those cases in appendix D.

### 3.4 Prioritized Training

Prioritized training refers to selection of most useful points for training in an online fashion with the aim of speeding up the training process. We consider the online batch selection scenario presented in Mindermann et al. (2022), where we only train on a selected 10% of the examples in each minibatch. Simple baselines for this task include selecting points with high loss or uniform random selection. It can be helpful to prioritize examples which are not yet learned (i.e. consistently correctly classified), but this can also select for mislabeled examples, which are common in large web-scraped datasets such as Clothing1M (Xiao et al., 2015). As noted by Mindermann et al. (2022), we need to find points which are *useful* to learn. Applying *MAP-D* in this context allows us to leverage training dynamics to identify such examples - we look for examples that are not already learned, but which still have training dynamics that resemble clean data:

```
training_score = (clean_score + (1. - correct_class_confidence)) / 2.     (9)
```

where `clean_score` is the probability of an example being clean (vs. noisy) according to the k-NN classifier described in Section 2.1.1. An example can achieve a maximum score of 1 under this metric when *MAP-D* predicts the example is clean, but the model assigns 0 probability to the correct label. Following Mindermann et al. (2022), we select 32 examples from each minibatch of 320. For (class-)balanced sampling, we also ensure that we always select at least 2 examples from each of the 14 possible classes, which significantly improves performance. Figure 6 shows the effectiveness of this approach vs. these baselines; we achieve a $\sim 10\mathrm{x}$ stepwise speedup over uniform random selection of examples.

We also report the original results from Mindermann et al. (2022) for reference which uses a different reporting interval. Mindermann et al. (2022) requires pretraining a separate model, and uses the prediction of that model to decide which points to prioritize for training. Our method on the other hand uses an online *MAP-D* trajectory scheme to decide whether an example is clean or noisy[2]. It is important to note that using balanced sampling with RHO Loss is likely to also improve performance for Mindermann et al. (2022).

### 3.5 Detection of Minority Group Samples

Minimizing average-case error often hurts performance on minority sub-groups that might be present in a dataset (Sagawa et al., 2019; 2020; Liu et al., 2021). For instance, models might learn to rely on spurious features that are only predictive for majority groups. Identifying minority-group samples can help detect and correct such issues, improving model fairness.

Previous works identify minority examples as those that are not already fit after some number of training epochs, and retrain from scratch with those examples upweighted (Liu et al.,

---

[2]We append the loss values of all examples in the batch to their learning curves before computing the assignments in order to ensure that examples can be correctly assigned even at the first epoch.

2021; Zhang et al., 2022). The number of epochs is treated as a hyperparameter; tuning it requires running the training process twice (without and then with upweighting) and evaluating on held-out known-to-be-minority examples. Instead of relying on the inductive bias that minority examples will be harder to fit, we apply *MAP-D* to find examples that match minority examples' training dynamics, and find this is much more effective method of identifying minority examples, see Figure 7. This avoids the costly hyperparameter tuning required by previous methods. Instead of just using 250 examples per probe category, we use the complete validation set in order to enable a fair comparison with JTT (Liu et al., 2021). Furthermore, these examples are not included as part of the training set in order to match the statistics of examples at test time.

## 4 RELATED WORK

We divide the related work into two major categories, starting from metadata inference which is the primary goal of our work, followed by potential interventions based on the discovered metadata. We provide a more holistic discussion of related work in Appendix F.

### 4.1 METADATA INFERENCE

Our work primarily relates to metadata inference. We consider metadata which is relational in nature. Individual efforts have been targeted towards different metadata properties in isolation, where an example is only ranked along one axis. Examples of such metadata is mislabeled examples (Arazo et al., 2019), typical/atypical examples (Brown et al., 2020; Jiang et al., 2020), difficult examples (Agarwal et al., 2021), minority-group examples (Liu et al., 2021; Sagawa et al., 2020; 2019; Zhang et al., 2022), points worth training (Mindermann et al., 2022), or domain identity in a domain generalization setting (Creager et al., 2021). *MAP-D* is a general method which enables metadata inference for different metadata categories in a consolidated framework leveraging the training dynamics of the network.

### 4.2 METADATA-SPECIFIC INTERVENTIONS

Once these metadata categories have been identified, different metadata specific interventions can be performed. Example of such interventions could be: correcting mislabeled examples present in a dataset (Arazo et al., 2019), using them only in a self-supervised training objective (Li et al., 2020a), or even completely ignoring them during training (Wang et al., 2018; Talukdar et al., 2021), upweighting or balancing training on atypical, or avoid memorization of noisy labels (Brown et al., 2020), upweighting minority-group samples to promote model fairness (Liu et al., 2021; Sagawa et al., 2020; 2019; Zhang et al., 2022), selectively training on the most important points in a batch (Mindermann et al., 2022), or perform group distributionally robust optimization using domain identities (Sagawa et al., 2019). We show that *MAP-D* can also be used to perform specific interventions once the metadata inference phase is completed.

## 5 CONCLUSION

We introduce the problem of *Metadata Archaeology* as the task of surfacing and inferring metadata of different examples in a dataset, noting that the relational qualities of metadata are of special interest (as compared to ordinary data features) for auditing, fairness, and many other applications. Metadata archaeology provides a unified framework for addressing multiple such data quality issues in large-scale datasets. We also propose a simple approach to this problem, *Metadata Archaeology via Probe Dynamics (MAP-D)*, based on the assumption that examples with similar learning dynamics represent the same metadata. We show that *MAP-D* is successful in identifying appropriate metadata features for data examples, even with no human labelling, making it a competitive approach for a variety of downstream tasks and datasets and a useful tool for auditing large-scale datasets. *MAP-D* can fail in scenarios where the training trajectories are not sufficiently distinct, or the probe suite is not correctly tailored for the task. We provide a detailed discussion of the limitations of our approach and future work in appendix G.

## 6 ACKNOWLEDGEMENTS

The authors would like to thank the SDS department at DFKI Kaiserslautern for their support with computing resources.

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

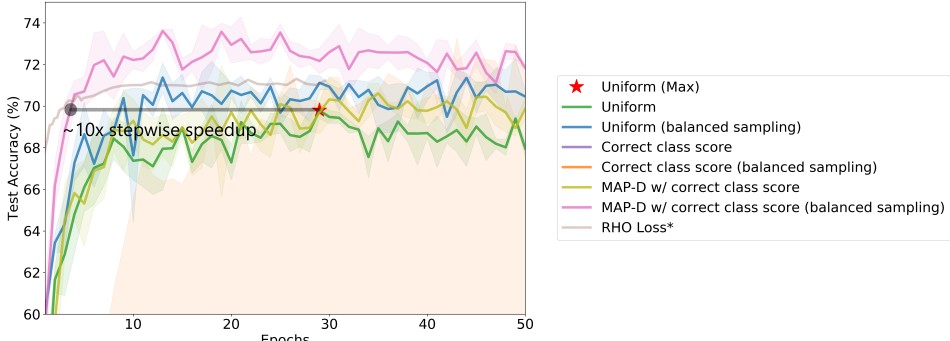

Figure 6: Results for score-based prioritization with *MAP-D* (**pink, lavender**) compared against simple baselines. Mean and standard deviation computed over 3 random runs. The correct class score baselines (**purple, orange**) both select examples with the highest loss, which lead to poor performance due to label noise. Uniform selection baselines (**blue, green**) perform quite well, but take longer to train; out method achieves almost a 10x stepwise training speedup. RHO Loss* baseline (**grey))** plots original results reported in Mindermann et al. (2022) while all other results use our implementation. While * use a different reporting interval, results remain comparable. The two methods are similar in training speed, but *MAP-D* achieves higher accuracy.

## A  EXPERIMENTAL DETAILS

In all experiments, we use variants of the ResNet architecture and leverage standard image classification datasets – CIFAR-10/100 and ImageNet. We train with SGD using standard hyperparameter settings: learning rate 0.1, momentum 0.9, weight-decay 0.0005, and a cosine learning rate decay. We achieve top-1 accuracies of 93.68% on CIFAR-10, 72.80% on CIFAR-100, and 73.94% on ImageNet.

**CIFAR-10/100**   To account for the smaller image size in this dataset, we follow standard practice and modify the models input layer to have stride 1 and filter size 3. We use a batch-size of 128 and train for 150 epochs. We use random horizontal flips and take a random crop of size $32 \times 32$ after padding the image using reflection padding with a padding size of 4 (He et al., 2016). For label noise correction experiments, we follow the experimental protocol of Arazo et al. (2019) with ResNet-18 where we train the model for 300 epochs with SGD and an initial learning rate of 0.1 decayed by a factor of 0.1 at the $100^{\text{th}}$ and $250^{\text{th}}$ epoch. A weight decay of 0.0001 is also applied.

**ImageNet**   We use a batch-size of 256 and train for 100 epochs. We apply center crop augmentation for testing as per the common practice (i.e. resize image to $256 \times 256$ and take the center crop of size $224 \times 224$) (He et al., 2016; NVIDIA, 2022).

**Waterbirds / CelebA**   We use the same model architecture and hyperparameters as Liu et al. (2021) in order to enable a fair and direct comparison. All experiments are based on the default ResNet-50 architecture. The Waterbirds models are trained for 300 epochs using SGD with an initial learning rate of 0.00001, and a high weight decay of 1.0. The model was early-stopped after the $60^{\text{th}}$ epoch for JTT (Liu et al., 2021). The CelebA models are trained for 50 epochs using SGD with an initial learning rate of 0.00001, and a high weight decay of 0.1. The model was early-stopped after the first epoch for JTT (Liu et al., 2021).

**Clothing1M**   We use the online batch selection protocol from Mindermann et al. (2022) where 32 examples are chosen from a large batch of 320 examples for training at each step. Following Mindermann et al. (2022), we use AdamW optimizer with default hyperparameters as in PyTorch (Paszke et al., 2019) and ImageNet pretrained ResNet-50. No learning rate decay is applied in this case.

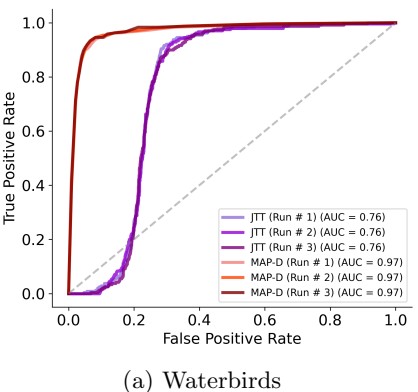
(a) Waterbirds

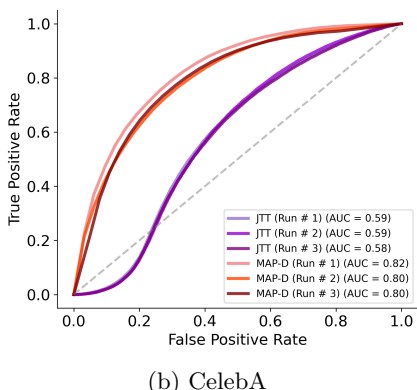
(b) CelebA

Figure 7: Demonstration of the effectiveness of *MAP-D* in detecting minority-group samples on two famous minority-group datasets with spurious correlations, compared to the detection performance of JTT (Liu et al., 2021) which relies on early-stopping. *MAP-D* achieves better or similar performance, without needing costly hyperparameter tuning or retraining.

## B    Examples considered to be mislabeled by *MAP-D*

Fig. 8 highlights images which *MAP-D* considers to be mislabeled from different ImageNet classes on the ImageNet train set. *MAP-D* is naturally very selective in considering examples to be mislabeled in contrast to conventional classifier-based approaches, which will assign an equal proportion of examples to the mislabeled set.

## C    Binary vs. Probabilistic Outputs in Label Correction

Arazo et al. (2019) used a convex combination of the labels weighted by the actual probability returned by their BMM model. As *MAP-D* returns probability estimates, this enabled leveraging label correction framework in the same way. However, the utility of the uncertainty estimates is not immediately apparent. Therefore, in order to gauge the utility of these uncertainty estimates, we used binary predictions (argmax) instead of the actual probabilities returned by *MAP-D*. The results are visualized in Fig. 9. It is clear from the figure that the model struggles significantly in coping with noise when being restricted to binary predictions, indicating that the uncertainty estimates provided by *MAP-D* enables the model to learn the correct label.

## D    Number of epochs before Label Correction

Arazo et al. (2019) segregated the complete 300 epochs of training into two phases: (i) pretraining phase where they train the model without label correction, and (ii) label correction phase for the rest of the 195 epochs during which they perform label correction. We observe that a relative strength of *MAP-D* is the ability to forgo such prolonged pretraining phase during training. We perform a simple experiment with a reduced number of pretraining epochs (10 instead of 105), leading to 290 epochs of training with label correction. These results are presented in Fig. 10, demonstrating that there is only negligible impact of pretraining schedule on *MAP-D* performance, while the performance of Arazo et al. (2019) is drastically impacted, specifically in no-noise and high-noise regimes.

## E    Probe Loss Distribution

We plot the probe loss distribution at specific epochs during training in Fig. 11. The figure also presents loss distribution from the validation/test probe examples where which are not directly used for the nearest neighbor classifier. Initially, the loss distribution for different probe categories is similar. However, as training progresses, the loss on some of the easier

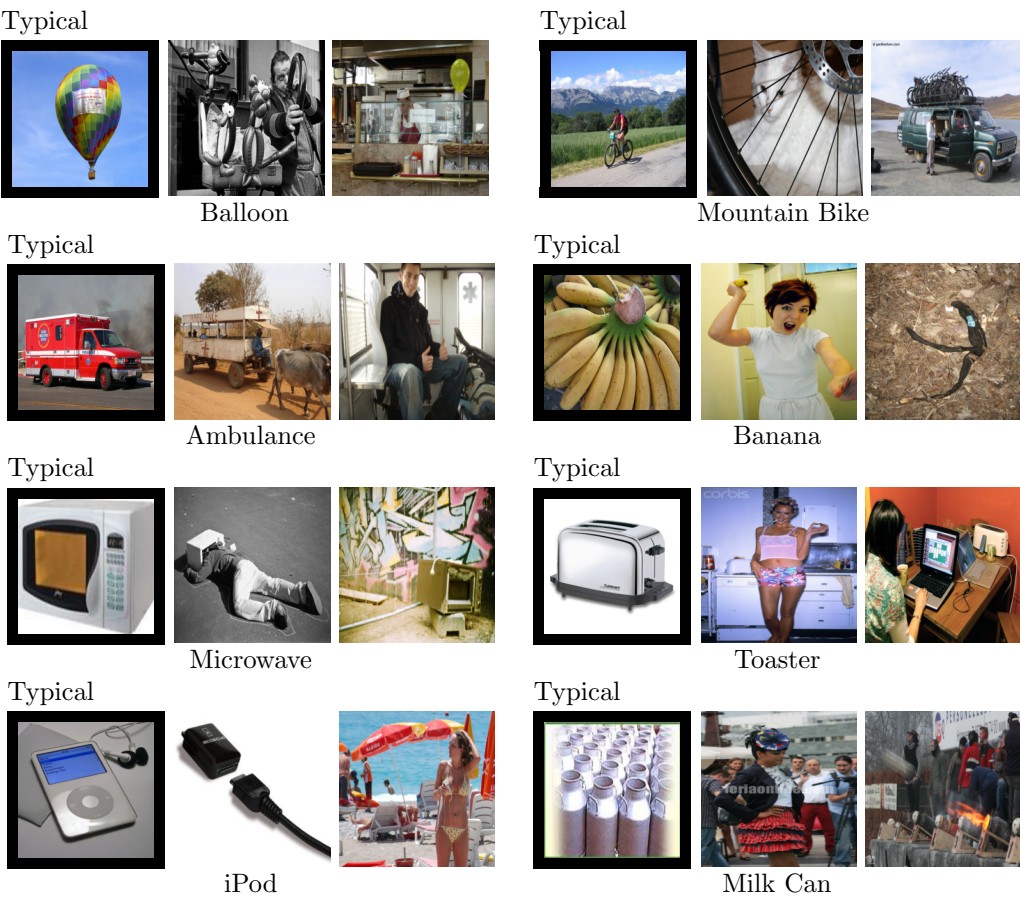

Figure 8: Examples surfaced through the use of *MAP-D* on ImageNet train set using the `Typical` probe (first image in each set, highlighted with **black border**) and `Random Output` probe (next two images). Sub-caption indicates the ground truth class. This showcases the utility of *MAP-D* for exploring a dataset, showing what the model considers typical for a class as well as uncovering potentially problematic examples.

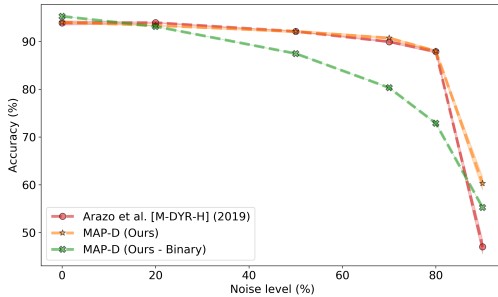

Figure 9: Ablation for label correction on CIFAR-10, where we use a binary prediction instead of probability estimates returned by *MAP-D*. This highlights the utility and effectiveness of the uncertainty estimates computed by *MAP-D*.

probe categories converges to zero. This illustrates that distinguishing examples based on loss distributions is possible, but difficult by just looking at one particular point in training. However, looking at the complete loss trajectory is sufficient to disambiguate the defined probe categories. We visualize the loss trajectories for all the defined probe categories in Fig. 12.

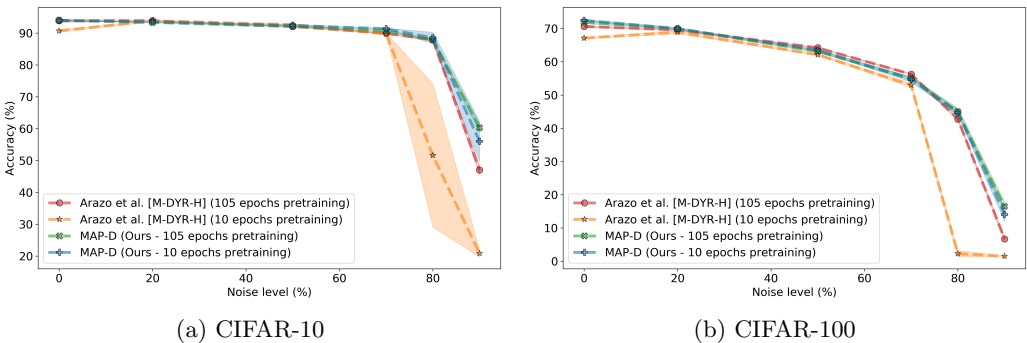

(a) CIFAR-10        (b) CIFAR-100

Figure 10: Comparison between pretraining schedules of 105 epochs (default value as set by Arazo et al. (2019)) and 10 epochs. Mean and standard deviation reported over 3 random runs. *MAP-D* is robust against changes in the number of pretraining epochs, while the method from Arazo et al. (2019) achieves slightly poorer performance in the low-noise setting and significantly poorer performance in the high-noise setting.

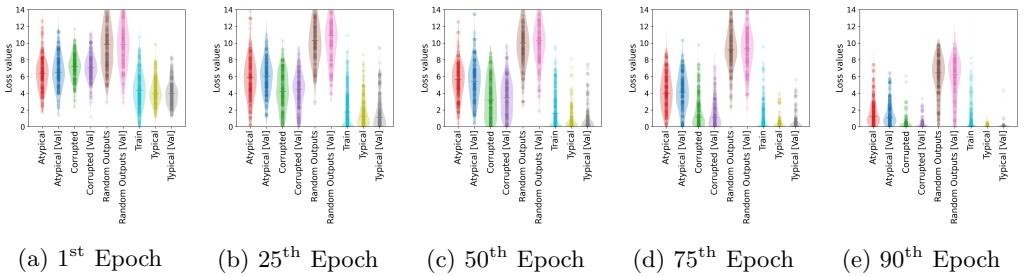

(a) 1$^{st}$ Epoch    (b) 25$^{th}$ Epoch    (c) 50$^{th}$ Epoch    (d) 75$^{th}$ Epoch    (e) 90$^{th}$ Epoch

Figure 11: Probe categories are distinguishable via learning dynamics of a ResNet-50 trained on ImageNet, validating the approach of *MAP-D*. For each of the probe categories, we plot the spread of losses at various epochs of training, including the validation/test probe examples. However, it is worth noting that deciding a single to distinguish these examples is difficult. On the other hand, using a full trajectory is sufficient for identification.

The figure also attempts to highlight that the loss distribution is similar for both training as well as validation/test probe examples from each of the probe categories.

## F   Related Work (Complete)

Many research directions focus on the properties of data and leveraging them in turn to improve the training process. We categorize and discuss each of these below.

**Monitoring per-example learning dynamics**   Kaplun et al. (2022) tracked training over each point individually in the dataset from multiple models in order to yield interesting insights into the structure of both models and data. Similarly, Rabanser et al. (2022) proposed a confident classification framework via monitoring the disagreement between different checkpoints throughout the training process and rejected predictions on samples with a significant disagreement. *MAP-D* is similar in spirit to this line of work, where we monitor loss values instead of softmax scores or model predictions on each individual example. However, *MAP-D* provides a more general framework to infer hidden metadata categories leveraging these training dynamics.

**Difficulty of examples**   Koh and Liang (2017) proposes influence functions to identify training points most influential on a given prediction. Work by Arpit et al. (2017); Li et al. (2020c); Feldman (2019); Feldman and Zhang (2020) develop methods that measure the degree of memorization required of individual examples. While Jiang et al. (2020) proposes

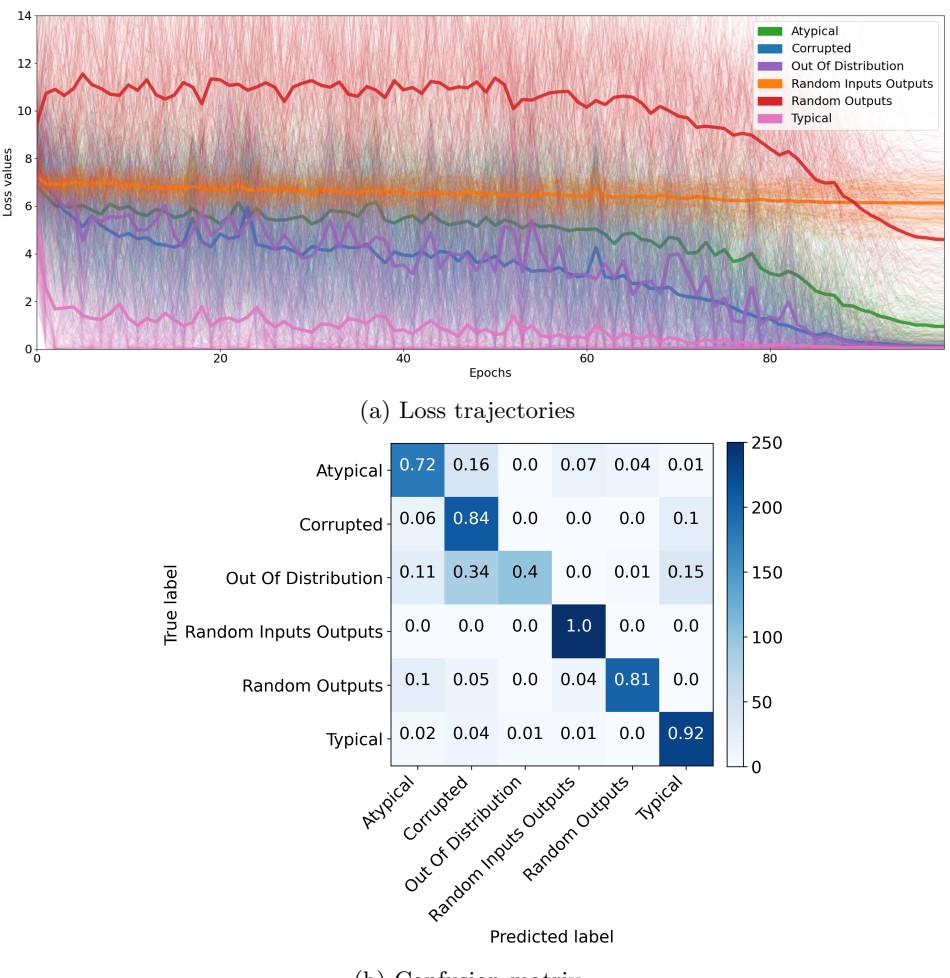

(a) Loss trajectories

(b) Confusion matrix

Figure 12: Sanity check showing performance of *MAP-D* on the probe suite test set with all probe categories using ResNet-50 on ImageNet, where we know the ground-truth metadata. **(a)** Solid line shows the mean learning curve while translucent lines are randomly sampled 250 individual trajectories for each probe category. Again, the separation of different probes is evident both in the dynamics over the course of training. **(b)** show confusion matrices between the true vs. predicted metadata features, demonstrating strong performance of the probes.

a consistency score to rank each example by alignment with the training instances, Carlini et al. (2019) considers several different measures to isolate prototypes that could conceivably be extended to rank the entire dataset. Agarwal et al. (2021) leverage variance of gradients across training to rank examples by learning difficulty. Further, Hooker et al. (2019) classify examples as challenging according to sensitivity to varying model capacity. In contrast to all these approaches that attempt to rank an example along one axis, *MAP-D* is able to discern between different sources of uncertainty without any significant computational cost by directly leveraging the training dynamics of the model.

**Coreset selection techniques**   The aim of these methods is to find prototypical examples that represent a larger corpus of data (Zhang, 1992; Bien and Tibshirani, 2012; Kim et al., 2015; Kim et al., 2016), which can be used to speed up training (Sener and Savarese, 2018; Shim et al., 2021; Huggins et al., 2017; Sorscher et al., 2022) or aid in interpretability of the model predictions (Yoon et al., 2019). *MAP-D* provides a computationally feasible alternate to identify and surface these coresets.

**Noisy examples**   A special case of example difficulty is noisy labels, and correcting for their presence. Arazo et al. (2019) use parameterized mixture models with two modes (for clean and noisy) fit to sample loss statistics, which they then use to relabel samples determined to be noisy. Li et al. (2020b) similarly uses mixture models to identify mislabelled samples, but actions on them by discarding the labels entirely and using these samples for auxiliary self-supervised training. These methods are unified by the goal of identifying examples that the model finds challenging, but unlike *MAP-D*, do not distinguish between the sources of this difficulty. The method from Arazo et al. (2019) particularly works under the assumption of uniform label noise. Other works such as Northcutt et al. (2021) tackles the case of non-uniform noise by estimating the noise matrix.

**Leveraging training signal**   There are several prior techniques that also leverage network training dynamics over distinct phases of learning (Achille et al., 2017; Jiang et al., 2020; Mangalam and Prabhu, 2019; Faghri et al., 2020; Agarwal et al., 2021). Notably, Pleiss et al. (2020) use loss dynamics of samples over the course of training, but calculate an Area-Under-Margin metric and show it can distinguish correct but difficult samples from mislabelled samples. In contrast, *MAP-D* is capable of inferring multiple data properties. Swayamdipta et al. (2020) computed the mean and variance of the model's confidence for the target label throughout training to identify interesting examples in the context of natural language processing. However, their method is limited in terms of identifying only easy, hard, or confusing examples. Our work builds upon this direction and can be extended to arbitrary sources of uncertainty based on defined probe suites leveraging loss trajectories.

**Adaptive training**   Adaptive training leverages training dynamics of the network to identify examples that are worth learning. Loss-based prioritization (Jiang et al., 2019; Katharopoulos and Fleuret, 2018) upweight high loss examples, assuming these examples are challenging yet learnable. These methods have been shown to quickly degrade in performance in the presence of even small amounts of noise since upweighting noisy samples hurts generalization (Hu et al., 2021; Paul et al., 2021). D'souza et al. (2021) motivate using targeted data augmentation to distinguish between different sources of uncertainty, and adapting training based upon differences in rates of learning. On the other hand, several methods prioritize learning on examples with a low loss assuming that they are more meaningful to learn. Recent work has also attempted to discern between points that are learnable (not noisy), worth learning (in distribution), and not yet learned (not redundant) (Mindermann et al., 2022). *MAP-D* can also be leveraged for adaptive training by defining the different sources of uncertainties of interest.

**Minority group samples**   The recent interest has been particularly towards finding and dealing with minority group samples to promote model fairness (Sagawa et al., 2019; 2020; Liu et al., 2021; Zhang et al., 2022; Nam et al., 2022). The dominant approach to deal with this problem without assuming access to group labels is to either pseudo-label the dataset using a classifier (Nam et al., 2022) or to train a model with early-stopping via a small validation set to surface minority group samples (Liu et al., 2021; Zhang et al., 2022). However, this setting only works for the contrived datasets where the model can classify the group based on the background. *MAP-D* leverages the population statistics rather than exploiting the curation process of the dataset to naturally surface minority group samples, which is scalable and applicable in the real-world.

## G   LIMITATIONS AND FUTURE WORK

*MAP-D* surfaces examples from the model based on the loss trajectories. This is based on a strong assumption that these loss trajectories are separable. It is possible that the learning curve for two set of probe categories exhibit similar behavior, limiting the model's capacity in telling them apart. In this case, the learning curve is no longer a valid discriminator between probes.

Furthermore, developing an appropriate probe-suite for a given task is non-trivial. As we use automated techniques, the effectiveness of the curated probe suite can be low for certain

applications. Furthermore, the automated techniques leveraging for designing the probe suite might not be applicable for a particular modality. In practice, designing a good and appropriate probe suite which elicits the right information from the model is a difficult task.

However, for good constructions of probe categories relying on global population statistics, we consider *MAP-D* to be a competitive and data-efficient method.

This work is focused on a computer vision setting; we consider an important direction of future work to be extending this to other domains such as speech or NLP.

