# OpenReview forum: "Metadata Archaeology: Unearthing Data Subsets by Leveraging Training Dynamics"
_ICLR.cc/2023/Conference — ICLR 2023 notable top 25%_

### Official Review · Reviewer_3RHf · 2022-10-23

**Confidence:** 3
**Correctness:** 3
**Technical Novelty And Significance:** 3
**Empirical Novelty And Significance:** 3
**Recommendation:** 8

**Clarity, Quality, Novelty And Reproducibility:**

The approach proposed in the paper appear to be novel. The paper is reasonably well-written and organized. However, it could be further improved by ...
(1) making Figure 3.a/b/c readable by using six or nine graphs instead the current three, so that the curves are not so "squashed" against each other;
(2) adding a paragraph to discuss in-depth the insights of what happens in Figure 3.d/e/f;
(3) expanding the last paragraph before "3.3" to a comprehensive discussion of the insight from Fig. 4;
(4) emphasizing earlier (beginning with the abstract!) MAP-D's impact on noise correction, prioritizing points for training, and identifying minority-group samples (in particular, the last two should be brought in the main paper, rather than relegated to appendices). The paper should find a way to summarize (in a compelling manner) across all 6 datasets, rather than just the two CIFARs. To save space in the main paper, the author(s) could shorten the introduction and move Figure 6 in an appendix (after all, its results could be summarize in one sentence, with the interested reader referred to the appropriate appendix).

**Strength And Weaknesses:**

The paper's main strengths are the novelty, simplicity, and effectiveness of MAP-D. It is also reasonably well-written and organized.

The paper's main weakness is the lack of an organic, real-world dataset on which to compare MAP-D against a high-quality, human-annotated test set. The probes described in section 2.2 (i.e., typical, atypical, random labels, random input & labels, and corrupted inputs) are intuitive and easy to compute w/o human annotations. However, understanding MAP-D's performance against a human-curated test set would make the paper significantly more compelling, while also highlighting its practical strengths and weaknesses in a real-world scenario.

**Summary Of The Paper:**

The paper introduces MAP-D, a novel approach to data archeology that leverages training dynamics to uncover a dataset's salient meta-data. In contrast to already existing approaches, MAP-D enables the simultaneous auditing of a dataset across multiple dimensions (e.g., typical data, corrupted inputs, etc).

**Summary Of The Review:**

The paper introduces a novel, simple, effective approach to an important problem: meta-data archeology. Overall, it is well written and organized, and it makes a strong empirical case. The paper could be further improved by (i) adding more in-depth discussions of the results, and (ii) further emphasize MAP-DF's impact on noise correction, prioritizing points for training, and identifying minority-group samples.

---

> ### Author Response · Authors · 2022-11-17
> **Response to reviewer 3RHf**
>
> We are thankful to the reviewer for a thorough, detailed and insightful review of our paper. We respond to each of the comments/questions/objections raised by the reviewer.
>
> 1. The paper's main weakness is the lack of an organic, real-world dataset on which to compare MAP-D against a high-quality, human-annotated test set. The probes described in section 2.2 (i.e., typical, atypical, random labels, random input & labels, and corrupted inputs) are intuitive and easy to compute w/o human annotations. However, understanding MAP-D's performance against a human-curated test set would make the paper significantly more compelling, while also highlighting its practical strengths and weaknesses in a real-world scenario.
> - Response: Thanks for raising this interesting point. We agree that an interesting extension to this work is to evaluate MAP-D where one of the probes is curated with human annotation. However, this process requires extreme care, as well as significant work.  Here, our focus was on probes that can be automatically curated. The reason for this was our interest in scalable methods for large datasets when human annotation is not feasible. However, we appreciate the reviewer’s suggestion and consider this to be a very natural avenue to be explored in future work.
>
> 2. Making Figure 3.a/b/c readable by using six or nine graphs instead the current three, so that the curves are not so "squashed" against each other
> - Response: Thank you for the useful feedback on this figure. We have incorporated the provided feedback to update the manuscript. We have cleaned up Fig. 3 (a-c) by replotting and moved the rest of the figures in Fig. 3 (d-h) to appendix.
>
> 3. Adding a paragraph to discuss in-depth the insights of what happens in Figure 3.d/e/f;
> - Response: Thank you for highlighting these missing details. As we moved these figures to the appendix, we had sufficient space to discuss them in detail, including what they are attempting to highlight.
>
> 4. Expanding the last paragraph before "3.3" to a comprehensive discussion of the insight from Fig. 4.
> - Response: Thank you for highlighting these missing details. We have attempted to elaborate on this in the paper at the end of section 3.3 given the limited amount of space.
>
> 5. Emphasizing earlier (beginning with the abstract!) MAP-D's impact on noise correction, prioritizing points for training, and identifying minority-group samples (in particular, the last two should be brought in the main paper, rather than relegated to appendices. The paper should find a way to summarize (in a compelling manner) across all 6 datasets, rather than just the two CIFARs. To save space in the main paper, the author(s) could shorten the introduction and move Figure 6 in an appendix (after all, its results could be summarize in one sentence, with the interested reader referred to the appropriate appendix).
> - Response: Thanks for this useful and important feedback regarding the paper organization. We have brought back the prioritized training and minority group detection experiments in the main paper while moving some unimportant parts to the appendix in line with the suggestions of all reviewers. We are thankful for these useful suggestions as we believe that these suggestions have significantly improved the overall organization of the paper. We have also moved Fig. 6 to the appendix and summarized the results in a single sentence in the main paper as the reviewer suggested.

---

### Official Review · Reviewer_Kiey · 2022-10-24

**Confidence:** 4
**Correctness:** 3
**Technical Novelty And Significance:** 3
**Empirical Novelty And Significance:** 3
**Recommendation:** 8

**Clarity, Quality, Novelty And Reproducibility:**

**Measuring Training Speed-Ups**

Speedups in units of steps or epochs do not consistently yield real-world (wall-clock) speedups. I think it would significantly strengthen the work if the authors demonstrated a wall-clock time improvement using their method for prioritized training, instead of just a step-wise speedup. At the very least, be specific about the units of speedup (e.g. "10x stepwise speedup" instead of "10x speedup")

**Clarity and Organization**

Related work section is thorough, but may want to include the recently-published Sorscher et al., 2022: Beyond neural scaling laws.

The text annotations in Figure 3 are difficult to read. I recommend enlarging them.

The main text makes it seem like Figures 5 and 6 are based on using the binary (argmax) method for label noise correction with MAP-D. It requires reading the appendix and comparing Appendix Figure 8 to Figures 6 and 6 to ascertain that the values in Figures 5 and 6 are obtained using the probabilistic method. I did not notice the probabilistic method mentioned in the main text at all. Please clarify this.

I think the noise correction correction protocol could be a little bit better explained in the main text. Specifically, does “number of epochs before noise correction” mean that the model is pretrained for N epochs (105 or 10 in the experiments here) in addition to the normal training duration, or that the total training duration is identical unchanged and the noise correction does not begin until N epochs into training?

The sample prioritization results are strong. Given that they’re one of the key results (and mentioned in the abstract and introduction), I think it undersells them to not describe the results, and to mention in passing at the beginning of the results section simply that they’re presented in the appendix. I understand space is limited, but I would suggest at least including a brief summary of the results in a dedicated subsection—it could be as little as one or two sentences.

I think some of the main text figures could be moved to the Appendix: specifically, Figure 3d-h, one of Figure 4a&b or Figure 4c&d, and Figure 6. This would allow moving the sample prioritization results and full related work section to the main text, as well as permitting more explanation and discussion of results.


**Strength And Weaknesses:**

**Strengths**

Simple method

Convincing results; method seems promising for multiple different applications

Well-written and mostly easy to follow

**Weaknesses**

Training speed-ups not measured in wall clock time

A few clarity issues, mostly regarding explanation of the noise correction methodology

Some of the more important content is in the Appendix, and some of what I would consider follow-up experiments are presented in the main text

**Summary Of The Paper:**

**Note: Score updated from 6 to 8 after authors' response.**

The authors propose a data sample annotation method they term Metatdata Archaeology via Probe Dynamics (MAP-D). In this method, one first creates data subsets with known properties (“probes”), such as corrupted, mislabeled, and atypical/typical samples. One then checks whether other samples have these properties by performing kNN classification on the time series of losses of unmodified data samples, with the time series of losses of the probes serving as the cluster labels. The authors apply MAP-D to six different image classification datasets and find that MAP-D performs similarly to SOTA baselines for identifying and correcting mislabeled examples, and can be used to speed up training by prioritizing training samples.

**Summary Of The Review:**

Straightforward method that shows promise for multiple different applications. I have a few suggestions to improve clarity and one to more convincingly measure a claimed training speed-up, but overall great work.

---

> ### Author Response · Authors · 2022-11-17
> **Response to reviewer Kiey (1/2)**
>
> We are thankful to the reviewer for spending his/her time reviewing our manuscript and for providing detailed and constructive feedback. We respond to each of the comments/questions/objections raised by the reviewer.
>
> 1. I think the noise correction protocol could be a little bit better explained in the main text. Specifically, does “number of epochs before noise correction” mean that the model is pretrained for N epochs (105 or 10 in the experiments here) in addition to the normal training duration, or that the total training duration is identical unchanged and the noise correction does not begin until N epochs into training?
> - Response: Thank you for asking to clarify this. Arazo et al. broke down the complete 300 epochs of training into two phases: (i) pretraining phase spanning 105 epochs where they train the model without label correction, and (ii) the label correction phase for the rest of the 195 epochs during which they perform label correction. Therefore, the total training duration is unchanged (300 epochs), but the noise correction does not begin until 105 epochs into training. We have updated the text to clarify this.
>
> 2. The text annotations in Figure 3 are difficult to read. I recommend enlarging them.
> - Response: Thank you for highlighting this. We were already considering the figure to be overly cluttered. We appreciate the feedback and have adjusted the figure as per the reviewer’s suggestion (including moving parts of the figure to the appendix).
>
> 3. The main text makes it seem like Figures 5 and 6 are based on using the binary (argmax) method for label noise correction with MAP-D. It requires reading the appendix and comparing Appendix Figure 8 to Figures 6 and 6 to ascertain that the values in Figures 5 and 6 are obtained using the probabilistic method. I did not notice the probabilistic method mentioned in the main text at all. Please clarify this.
> - Response: Thanks for highlighting this. We have highlighted this in the text to make it clear that we use the probabilistic version of MAP-D for label correction experiments, and refer to the appendix for more details.
>
> 4. Some of the more important content is in the Appendix, and some of what I would consider follow-up experiments are presented in the main text.
> - Response: Thanks for highlighting this. Our manuscript is much improved because of these suggestions. We attempted to balance the need for details within each experiment and the space required to do so, within the limitation of 9 pages. We found the detailed feedback on the organization very helpful and have updated the manuscript to accommodate this feedback in terms of writing, moving some sections back in, and moving some figures to the appendix. We now have details regarding prioritized training and minority group detection within the main paper and left the corresponding figures in the appendix. We hope that the new organization will be significantly more interesting for the readers.

---

> > ### Comment · Reviewer_Kiey · 2022-11-17
> > **Thank you for your response! I have updated my score**
> >
> > I appreciate your thorough response. I have updated my score from a 6 to an 8 accordingly.

---

> > > ### Author Response · Authors · 2022-11-17
> > > **Thank you for helping us improve the paper**
> > >
> > > We are thankful to the reviewer for helping us improve the quality of our work, as well as for increasing their score.

---

> ### Author Response · Authors · 2022-11-17
> **Response to reviewer Kiey (2/2)**
>
> 5. I think some of the main text figures could be moved to the Appendix: specifically, Figure 3d-h, one of Figure 4a&b or Figure 4c&d, and Figure 6. This would allow moving the sample prioritization results and full related work section to the main text, as well as permitting more explanation and discussion of results.
> - Response: Thanks for indicating this. We agree. We have updated the paper by moving the suggested figures to the appendix and moving the prioritized training and minority group detection experiments to the main text.
>
> 6. The sample prioritization results are strong. Given that they’re one of the key results (and mentioned in the abstract and introduction), I think it undersells them to not describe the results, and to mention in passing at the beginning of the results section simply that they’re presented in the appendix. I understand space is limited, but I would suggest at least including a brief summary of the results in a dedicated subsection—it could be as little as one or two sentences.
> - Response: We would like to thank the reviewer for this feedback. Based on feedback from the reviewer, as well as other reviewers regarding the organization, we have included the textual description of our results on prioritized training and minority group detection after making space through moving figures as mentioned by the reviewer, while we left the figures in the appendix.
>
> 7. Training speed-ups not measured in wall clock time
> - Response: Thanks for highlighting this. We report the same metric as reported in the original paper we compared against on prioritized training (Mindermann et al. 2022). We do agree that reporting wall clock time is interesting. The reason why we refrain from reporting wall-clock time is primarily because of the dependence on implementation details including hardware. We consider these optimizations to be outside the scope of this work. Our aim is to just demonstrate that MAP-D, despite being a very general approach, also works particularly well for specific tasks such as label correction or prioritized training.
>
> 8. At the very least, be specific about the units of speedup (e.g. "10x stepwise speedup" instead of "10x speedup")
> - Response: Thanks for the suggestion. We have updated the paper to clarify this i.e. “10x stepwise speedup“ instead of “10x speedup”.

---

### Official Review · Reviewer_KmwB · 2022-10-24

**Confidence:** 5
**Clarity, Quality, Novelty And Reproducibility:** See Strength And Weaknesses
**Correctness:** 3
**Technical Novelty And Significance:** 3
**Empirical Novelty And Significance:** 3
**Recommendation:** 5

**Strength And Weaknesses:**

Pros:
1. The motivation is clear.
2. The paper is well-written and organized.
Cons:
1. The main contributions are not clear.
2. Some related works are missing, e.g., 3D Face Reconstruction from A Single Image Assisted by 2D Face Images in the Wild.

**Summary Of The Paper:**

This paper proposes Metadata Archaeology, a unifying and general framework for uncovering latent metadata categories.
This paper introduces and validate the approach of Metadata Archaeology via Probe Dynamics (MAP-D): leveraging the training dynamics of curated data subsets called probe suites to infer other examples’ metadata.
This paper shows how MAP-D could be leveraged to audit large-scale datasets or debug model training, with negligible added cost. This is in contrast to prior work which presents a siloed treatment of different data properties.
The authors use MAP-D to identify and correct mislabeled examples in a dataset. Despite its simplicity, MAP-D is on-par with far more sophisticated methods, while enabling natural extension to an arbitrary number of modes.
The authors show how to use MAP-D to identify minority group samples, or surface examples for data-efficient prioritized training.

**Summary Of The Review:**

See Strength And Weaknesses

---

> ### Author Response · Authors · 2022-11-14
> **Response to reviewer KmwB**
>
> We thank reviewer KmwB for their input on our work. We are constrained by the lack of detail in the review to fully address the perceived limitations which resulted in this low score by the reviewer. We welcome additional clarification, and we respond to the limited questions/objections raised by the reviewer below.
>
> 1. The main contributions are not clear.
> - Response: We state our contributions at the end of the introduction section of the paper. We take this opportunity to restate our main contributions here: we introduce a new framework called MAP-D to infer characteristics of data distribution along multiple different axes simultaneously, while previous work provided siloed treatments of each of these metadata properties. We demonstrated the effectiveness of our framework on data auditing, which is a primary goal of our work. Furthermore, we show that MAP-D can work on metadata-specific interventions such as label correction, prioritized training, or even identification of minority group points in the dataset. Having additional details from the reviewer about the lack of clarity around contributions would help us be able to address these concerns during the rebuttal period.
>
> 2. Some related works are missing, e.g., 3D Face Reconstruction from A Single Image Assisted by 2D Face Images in the Wild.
> - Response: Thanks for highlighting this. We looked at the reference, and found no clear relevance to our work. We would welcome any clarification from the reviewer. We are happy to update our work to reflect any missing citations.
>
> We hope the reviewer will consider raising their score or provide additional guidance to justify their review such that we can respond adequately within the rebuttal period.

---

### Official Review · Reviewer_o59Z · 2022-10-25

**Confidence:** 4
**Correctness:** 3
**Technical Novelty And Significance:** 3
**Empirical Novelty And Significance:** 3
**Recommendation:** 8

**Clarity, Quality, Novelty And Reproducibility:**

The paper is clear and very well written. The paper solves a novel problem. It seems to be reproducible if the code is released upon acceptance of the paper.

**Strength And Weaknesses:**

*Strengths*:
1. The paper proposes, to the best of my knowledge, the first method which jointly infers multiple meta-data features of a dataset.
2. Simplicity: The method is simple and intuitive.
3. Paper is very well-written.

*Weaknesses*:
1. While I do appreciate the simplicity of the methods, it is hard to believe that the method performs on par with state-of-the-art label correction methods. I believe this can only be the case when the label corruptions occur uniformly at random, which might very well not be the case, as shown by many recent paper on noisy label correction. Moreover, the papers that the authors compare with are not the state-of-the-art since methods like Confident learning proposed later outperform the methods the authors compare with.
2. The method does not seem to detect out-of-distribution samples very well (Fig. 4 (d)). I would like the authors to explain the underwhelming performance.
3. It would be nice and instructive to discuss failure of the method in detecting these meta-data features. Like the authors mention in the limitations, the assumption that the learning dynamics.
4. The authors mention experiments on two different kinds on ResNet models, but (1) I do not see results from both the models, (2) experimenting with different model architectures might make for a stronger experimental result, showing the method's applicability beyond ResNet architectures.

*Questions*
1. How are out-of-distribution probes generated?

**Summary Of The Paper:**

The paper provides a method to infer various meta-data features of datasets such as typical/atypical data points, noisy labels, noisy and out-of-distribution data points.

**Summary Of The Review:**

I like the paper, I think it is novel, simple, intuitive, well-written with strong experimental results, but can benefit from some more experiments and clarifications.

---

> ### Author Response · Authors · 2022-11-18
> **Response to reviewer o59Z**
>
> We thank the reviewer for a detailed and insightful review. We appreciate the positive feedback from the reviewer, including the contributions being significant and the work being well-written with strong experimental results. We enjoyed reading the comments from the reviewer, as the points made were really insightful and interesting. We respond to each of the questions/objections raised by the reviewer below:
>
> 1. While I do appreciate the simplicity of the methods, it is hard to believe that the method performs on par with state-of-the-art label correction methods. I believe this can only be the case when the label corruptions occur uniformly at random, which might very well not be the case, as shown by many recent papers on noisy label correction. Moreover, the papers that the authors compare with are not the state-of-the-art since methods like Confident learning proposed later outperform the methods the authors compare with.
> - Response: Thank you for this interesting question. The reviewer was correct in highlighting that we compare MAP-D in a setting that assumes uniform noise distribution. Our goal here is to illustrate the applicability of a general framework on a range of metadata-specific interventions. Hence, we focused on one type of noise distribution to illustrate the versatility of our approach. Indeed, mitigation strategies for uniform and non-uniform noise correction are typically different, with non-uniform first requiring estimation of the uncertainty matrix for different classes. We have updated the manuscript to clarify that we evaluate in a uniform noise setting.
>
> 2. How are out-of-distribution probes generated? The method does not seem to detect out-of-distribution samples very well (Fig. 4 (d)). I would like the authors to explain the underwhelming performance.
> - Response: Thank you for raising this interesting question. OOD probes were generated using a specific edge filter to obtain sketch-like images. During the rebuttal period, we ran an ablation study to better understand the performance of the OOD variant. We observed that if corrupted/atypical probes are removed, OOD detection using MAP-D improves. This was a valuable question because this highlights that our OOD definition is in some ways a type of “atypical” instance – hence both were easily conflated as probes. It is hard to find a preexisting OOD dataset with the same set of classes, but with instances that cannot be simply considered a type of atypical/corrupted instances, as well as exhibiting distinct learning dynamics.
>
> 3. It would be nice and instructive to discuss failure of the method in detecting these meta-data features. Like the authors mention in the limitations, the assumption that the learning dynamics.
> - Response: Thank you for this interesting feedback. We agree that it would be interesting to discuss them. The reviewer has already highlighted one limitation by pointing to the relatively poor performance on OOD probes, which can occur if probe categories are too semantically close to each other. Since this has instructive value as correctly pointed out by the reviewer, we added a new section in the appendix (appendix G) to cover this in greater detail. Although we would prefer to cover this in more detail directly in the paper, we had to bring back other sections from appendix in the paper based on comments from all reviewers.
>
> 4. The authors mention experiments on two different kinds on ResNet models, but (1) I do not see results from both the models, (2) experimenting with different model architectures might make for a stronger experimental result, showing the method's applicability beyond ResNet architectures.
> - Response: Thanks for highlighting this insufficiency of details. The experiments on label correction are based on ResNet-18, while all other results are based on ResNet-50. We mentioned this in appendix A, but we have also updated the end of section 2 to include these details and refer to appendix A for more information.

---

### Decision · Program_Chairs · 2023-01-20

**Decision:**

Accept: notable-top-25%

**Justification For Why Not Higher Score:**

Reviewers have brought up some issues highlighting limits in empirical evaluation scope and design.


**Justification For Why Not Lower Score:**

This is a good, novel and potentially useful work with a good impact potential to enable comprehensive meta-learning and related tasks.
The reviewers' concerns are not major and some of them have been already addressed by the authors via the rebuttal.

**Metareview: Summary, Strengths And Weaknesses:**

This is a well written paper presenting a novel contribution accompanied with a well designed set of convincing experiments (with some room for improvements in he empirical scope and process). It has been assessed by four knowledgeable reviewers, three of whom recommend a straight accept an one rates the work as marginally rejectable. Based on my review, I would like to side with the majority of my colleagues and recommend acceptance of the paper in its current form. The authors have actively communicated with the reviewers during the rebuttal period and they seem to have addressed most of the pressing concerns already.

**Note From Pc:**

if the above contains the word "oral" or "spotlight" please see: "oral" presentation means -> notable-top-5% and "spotlight" means -> notable-top-25%. As stated in our emails, we are disassociating presentation type from AC recommendations